# Carbon Nanodisks Decorated with Guanidinylated Hyperbranched Polyethyleneimine Derivatives as Efficient Antibacterial Agents

**DOI:** 10.3390/nano14080677

**Published:** 2024-04-13

**Authors:** Kyriaki-Marina Lyra, Ioannis Tournis, Mohammed Subrati, Konstantinos Spyrou, Aggeliki Papavasiliou, Chrysoula Athanasekou, Sergios Papageorgiou, Elias Sakellis, Michael A. Karakassides, Zili Sideratou

**Affiliations:** 1Institute of Nanoscience and Nanotechnology, National Center for Scientific Reasearch “Demokritos”, Aghia Paraskevi, 15310 Athens, Greece; k.lyra@inn.demokritos.gr (K.-M.L.); i.tournis@inn.demokritos.gr (I.T.); mossubrati@gmail.com (M.S.); a.papavasiliou@inn.demokritos.gr (A.P.); c.athanasekou@inn.demokritos.gr (C.A.); s.papageorgiou@inn.demokritos.gr (S.P.); or e_sakel@phys.uoa.gr (E.S.); 2Department of Material Science & Engineering, University of Ioannina, 45110 Ioannina, Greece; konstantinos.spyrou1@gmail.com (K.S.); mkarakas@uoi.gr (M.A.K.); 3Physics Department, Condensed Matter Physics Section, National and Kapodistrian University of Athens, Panepistimiopolis, Zografos, 15784 Athens, Greece

**Keywords:** carbon-based nanomaterials, guanidinium groups, hyperbranched dendritic polymers, antibacterial properties

## Abstract

Non-toxic carbon-based hybrid nanomaterials based on carbon nanodisks were synthesized and assessed as novel antibacterial agents. Specifically, acid-treated carbon nanodisks (oxCNDs), as a safe alternative material to graphene oxide, interacted through covalent and non-covalent bonding with guanidinylated hyperbranched polyethyleneimine derivatives (GPEI5K and GPEI25K), affording the oxCNDs@GPEI5K and oxCNDs@GPEI25K hybrids. Their physico-chemical characterization confirmed the successful and homogenous attachment of GPEIs on the surface of oxCNDs, which, due to the presence of guanidinium groups, offered them improved aqueous stability. Moreover, the antibacterial activity of oxCNDs@GPEIs was evaluated against Gram-negative *E. coli* and Gram-positive *S. aureus* bacteria. It was found that both hybrids exhibited enhanced antibacterial activity, with oxCNDs@GPEI5K being more active than oxCNDs@GPEI25K. Their MIC and MBC values were found to be much lower than those of oxCNDs, revealing that the GPEI attachment endowed the hybrids with enhanced antibacterial properties. These improved properties were attributed to the polycationic character of the oxCNDs@GPEIs, which enables effective interaction with the bacterial cytoplasmic membrane and cell walls, leading to cell envelope damage, and eventually cell lysis. Finally, oxCNDs@GPEIs showed minimal cytotoxicity on mammalian cells, indicating that these hybrid nanomaterials have great potential to be used as safe and efficient antibacterial agents.

## 1. Introduction

In the recent past, infection with pathogenic bacteria became a global health burden due to emerging and drug-resistant bacteria defying clinical treatment. Many attempts have been made to investigate the mechanism of antibiotic resistance to address not only drug efficiency but also drug toxicity [1]. Nowadays, the development of new antibiotic-free antibacterial agents has become mandatory in order to overcome antibiotic resistance [2,3]. Graphene-based nanomaterials have been proposed as promising candidate systems that have the potential to fight diseases caused by both Gram-positive and Gram-negative bacteria [4,5]. Although the precise antibacterial mechanism of these nanomaterials remains controversial [6,7], the most predominant mechanisms proposed in the literature are related to the physical interaction of the sharp edges of graphenes with bacterial membranes, the generation of oxidative stress and the wrapping of bacterial cells by graphene sheets [4,5]. Additionally, it has been found that various other parameters, such as size, thickness and shape of the sheets, surface modification, agglomeration tendency and aqueous dispersibility as well as type of bacteria, etc., can influence the antibacterial activity of graphene-based nanomaterials [7,8].

Among these nanomaterials, graphene oxide’s (GO) antibacterial properties have been extensively studied in the last few decades, against both Gram-positive and Gram-negative bacteria. Even though the results of these studies are diverse due to the use of a variety of experimental methods, data presentation, bacteria strains, type and GO concentration, etc., it has been well-documented that GO exhibits a time and dose-dependent antibacterial effect on both Gram-positive and Gram-negative bacteria, attributed to mechanisms such as the generation of oxidative and membrane stress, bacteria entrapment into GO aggregations, etc. [9]. Specifically, the sharp edges of GO combined with the oxygen-containing groups that promote interactions with bacteria can damage bacterial membranes and walls, resulting in cell death without any intracellular mechanism being followed [10,11]. In addition, GO can produce ROS, generating oxidative stress that leads to DNA damage, bacterial dysfunction and finally bacterial death [12]. Furthermore, GO has the ability to form aggregates with bacteria [13], so the later are entrapped in them and cannot proliferate due to the inhibition of nutrient uptake, which ultimately leads to cell death [14]. To enhance these properties as well as to reduce the tendency to aggregate due to strong inter-plane interactions, various functionalization strategies have been proposed, including GO surface modification with polymers such as chitosan [15,16], poly-N-vinyl carbazole [17], poly-l-lysine [18], polyhexamethylene guanidine hydrochloride [16,19], polyethyleneimine [20], etc., through either covalent (chemical) or non-covalent (physical sorption) synthetic routes. Fan et al. [20] prepared a polyethyleneimine-modified graphene oxide via single-step synthesis and proved that this derivative exhibited enhanced aqueous dispersibility and antibacterial activity only on Gram-posotive methicillin-resistant *Staphylococcus aureus* and not on Gram-negative *Escherichia coli* bacteria. Li et al. [19] synthesized a multifunctional GO derivate, bearing polyethylene glycol chains and polyhexamethylene guanidine hydrochloride with enhanced aqueous dispersibility and antibacterial properties against Gram-negative and Gram-positive bacteria. Both of the above-mentioned GO derivatives can efficiently interact with bacterial membranes due to their improved aqueous colloidal stability and the presence of amino or guanidinium groups on their surface, resulting in the rupture or deformation of cell walls and membranes, and therefore in cell damage. Driven by these results, in the present study, two derivatives of hyperbranched polyethyleneimine functionalized with guanidium groups were synthesized and subsequently used to modify oxidized carbon nanodisks (oxCNDs), a novel member of the carbon-based family that has an analogous chemical structure to GO.

Carbon nanodisks (CNDs) [21] are ultra-thin (10–30 nm), quasi two-dimensional graphene-based particles with a diameter of 0.8–3 µm, produced from the pyrolysis of hydrocarbons following the Pyrolytic Kværner Carbon Black & H_2_ process [22] after post-treatment at 2700 °C under an argon atmosphere [23]. This procedure yields mainly carbon nanodisks (~75%) together with carbon nanocones (~20%). Due to their interesting properties, CNDs could represent an attractive alternative to bulk graphite [24]. However, just as all graphene-based nanomaterials, CNDs exhibit low dispersibility in aqueous media due to their high tendency to form strong van der Waals and π-π interactions, preventing their use in several applications. A means to overcome this obstacle is chemical oxidation based on the Staudenmaier method [25], commonly applied for the oxidation of graphite [26,27]. The resulting oxidized CNDs contain various oxygen-containing groups such as carboxyl, epoxy and hydroxyl groups on their surfaces, as graphene oxide has [28]. Comparing the structural characteristics of oxCNDs with those of GO, oxCNDs show a significantly more narrow size distribution combined with a well-defined disk-like shape [29,30], a lower ratio of C:O atoms due to the larger number of oxygen-containing groups on its surface and a higher content of functional groups located mainly at the discs edges that can be further modified [28]. Another crucial advantage is the low toxicity of oxCNDs, probably due to their disk-like structure [28], unlike GO, which is known to exhibit cytotoxicity against mammalian cells [31,32,33,34]. Thus, oxCNDs can be considered more advantageous materials than GO and in this study, they are proposed as a GO alternative.

In this context, in the present study, for the first time, oxidized carbon nanodisks were functionalized with guanidinylated hyperbranched polyethyleneimine derivatives (GPEIs). Thus, two guanidinylated derivatives of hyperbranched polyethyleneimine (PEI) with molecular weights of 5000 and 25,000 Da (GPEI5K or GPEI25K, respectively) were synthesized and subsequently interacted through covalent and non-covalent bonds affording novel hybrid materials (oxCNDs@GPEI5K and oxCNDs@GPEI5K) with enhanced aqueous dispersibility. After physico-chemical characterization of the as-prepared nanomaterials using a variety of techniques (XPS, FTIR, Raman, NMR, SEM, TEM, etc.), their antibacterial performance was evaluated against Gram-negative bacteria (*Escherichia coli*) and Gram-positive bacteria (*Staphylococcus aureus*), while their cytocompatibility was evaluated on mammalian cell lines.

## 2. Materials and Methods

### 2.1. Chemicals and Reagents

Carbon nanodisks were purchased from Strem Chemicals, Inc. (Bischheim, France). Hyperbranched polyethyleneimine (PEI) of 5,000 Da (PEI5K, Lupasol^®^ G100, water-free, 99%) and 25,000 Da molecular weight (PEI25K, Lupasol^®^ WF, water-free, 99%) was kindly donated by BASF (Ludwigshafen, Germany). 1*H*-Pyrazole-1-carboxamidine hydrochloride, *N*,*N*-diisopropylethylamine (DIPEA), thiazolyl blue tetrazolium bromide (MTT), sodium chloride (NaCl), tryptic soy broth (TSB), resazurin, glutaraldehyde (solution, 25%) and sodium cacodylate were purchased from Sigma-Aldrich Ltd. (Poole, UK). Dulbecco’s Modified Eagle Medium (DMEM), low glucose with phenol red, fetal bovine serum (FBS), penicillin/streptomycin, L-glutamine, phosphate buffer saline (PBS) and trypsin/EDTA were obtained from Biowest (Nuaillé, France). Peptone from Casein was obtained from AppliChem GmbH (Darmstadt, Germany), while agar and lambda broth (LB) were purchased from MP Biomedicals (Illkirch, France). High purity solvents such as *N*,*N*-dimethylformamide (DMF, anhydrous, 99.8%), ethanol (99.9%), methanol (≥99.8%) and isopropanol (99.8%) were obtained from Merck KGaA (Calbiochem^®^, Darmstadt, Germany).

### 2.2. Preparation of GPEI and oxCNDs

Guanidinylated PEI derivatives (GPEIs) with a nominal 100% substitution degree of primary amino groups were synthesized following a simple guanidinylation reaction as described in our previous publications [28,35]. Briefly, PEI5K or PEI25K (0.01 mmol) dissolved in DMF (10 mL) was added to a DMF solution (10 mL) containing 0.4 or 2 mmol 1*H*-Pyrazole-1-carboxamidine hydrochloride and 0.8 mmol or 4 mmol DIPEA, respectively. The mixture was stirred at room temperature in an inert atmosphere for 24 h. The final products, GPEI5K and GPEI25K, were received after precipitation in diethyl ether, dried and finally obtained after dialysis against deionized water and lyophilization. The successful introduction of guanidinium moieties and the degree of functionalization were established by ^1^H and ^13^C NMR spectroscopy, amounting to substitution degrees of 98% and 95% of the PEI primary amino groups for GPEI5K and GPEI25K, respectively.

Oxidized CNDs were prepared following a modified Staudenmaier method as previously described [28]. In brief, CNDs (300 mg) were dispersed in a mixture of 12 mL H_2_SO_4_ (95–97%) and 6 mL HNO_3_ (65%), while being cooled in an ice-water bath. The mixture was stirred for 30 min and after that, 6 g potassium chlorate in powder form was partially added under vigorous stirring and cooling in an ice-water bath. The reaction was quenched after 18 h by pouring the mixture into 200 mL distilled water and the final oxCNDs were obtained after washing with water four times and lyophilization.

### 2.3. Preparation of GPEI-Functionalized oxCNDs

The functionalization of oxCNDs with guanidinylated hyperbranched polyethyleneimine derivatives was performed following an analogous method previously described [35,36] In brief, 150 mg oxCNDs were dispersed in 50 mL distilled H_2_O with the aid of ultrasonication for 30 min, applying a Hielscher UP200S high intensity ultrasonic processor (Hielscher Ultrasonics GmbH, Teltow, Germany) coupled with a standard sonotrode (3 mm tip-diameter) at 50% amplitude and 0.5 cycles/s. Then, the pH value of the dispersion was adjusted to ~9 by the addition of NaOH solution (0.5 M) and left under stirring for 24 h, at room temperature. Subsequently, 50 mL of an aqueous solution of GPEI5K or GPEI25K (60 mg/mL) was added to the abovementioned oxCND dispersion. The reaction mixture was left under continuous stirring for 48 h, at room temperature. The final products, oxCNDs@GPEI5K or oxCNDs@GPEI25K, were obtained after centrifugation at 20,000× *g*, washing with water until the pH of the supernatant reached the value of 6.5 and lyophilization.

### 2.4. Characterization of GPEI-Functionalized oxCNDs

^1^H spectra were recorded at 25 °C on a Bruker Avance DRX spectrometer operating at 500 MHz (Bruker Biospin, Rheinstetten, Germany), using D_2_O as a solvent. The polymer content of GPEI-functionalized oxCNDs was determined by ^1^H NMR using maleic acid as an internal standard [37]. FTIR spectra were acquired on a Nicolet 6700 spectrometer (Thermo Scientific, Waltham, MA, USA) equipped with a Specac Quest ATR with a diamond crystal (Specac Ltd., London, UK) at 4 cm^−1^ resolution. Next, 100 μL of a methanol solution of oxCNDs@GPEIs (0.5 mg/mL) was deposited on the diamond element and the solvent was evaporated under a steam of nitrogen to produce a thin film. In total, 128 scans were collected and the signal was averaged. Raman spectra were recorded on a RM 1000 Renishaw micro-Raman system ((Renishaw, Wotton-under-Edge, England) using a laser excitation line at 532 nm (Nd-YAG) in the range of 400–2000 cm^−1^. X-ray photoelectron spectroscopy (XPS) measurements were carried out under an ultrahigh vacuum at a base pressure of 4 × 10^−10^ mbar using a SPECS GmbH spectrometer equipped with a monochromatic MgKa source (*hv* = 1253.6 eV) and a Phoibos-100 hemispherical analyzer (Berlin, Germany). The spectra were collected under normal emission and energy resolution was set to 1.16 eV to minimize the measuring time. Spectral analysis included a Shirley background subtraction and peak deconvolution employing mixed Gaussian–Lorentzian functions, in a least squares curve-fitting program (WinSpec) developed at the Laboratoire Interdisciplinaire de Spectroscopie Electronique, University of Namur, Belgium. All binding energies were referenced to the C1s core level of the photoemission line at 284.6 eV. Thermogravimetric analyses (TGA) were carried out on a Setaram SETSYS Evolution 17 analyzer (SETARAM Instrumentation, Caluire, France) at a 5 °C/min heating rate under air atmosphere. X-ray diffraction (XRD) data were obtained using a system consisting of a Rigaku RUH3R rotating anode generator operating at 50 kV and 100 mA, producing a beam of λ = 1.5416 Å, Kα line of Cu and an R-AXIS IV image plate (Rigaku Co., Tokyo, Japan). The samples in powder form were placed in Lindemann capillaries (Hilgenberg-Mark tubes of 1 mm inner diameter). Scanning electron microscopy (SEM) images were recorded using a Jeol JSM 7401F Field Emission Scanning Electron Microscope (JEOL USA Inc., Peabody, MA, USA) equipped with Gentle Beam mode. Transmission electron micrographs were taken using an FEI Talos F200i field-emission (scanning) transmission electron microscope (Thermo Fisher Scientific Inc., Waltham, MA, USA) operating at 200 kV, equipped with a 100 mm^2^ X-Flash 6|T windowless energy-dispersive spectroscopy microanalyzer (Bruker, Hamburg, Germany). For sample preparation, a drop of oxCNDs@GPEIs aqueous dispersion (0.1 mg/mL) was casted on a copper TEM grid covered with a thin carbon layer and left to evaporate. Elemental analysis (EA) was executed by a Perkin Elmer 240 CHN elemental analyzer (Perkin Elmer, Waltham, MA, USA). *ζ*-potential measurements of GPEI-functionalized oxCND aqueous dispersions (0.2 mg/mL) were performed employing a ZetaPlus (Brookhaven Instruments Corp., Long Island, NY, USA). For these experiments, each dispersion (800 μL) was diluted with an equal volume of water, ten measurements were collected and the results were averaged.

### 2.5. Assessment of the Antibacterial Properties

#### 2.5.1. Test Microorganisms

*Escherichia coli* strain DH5α (*E. coli*) and *Staphylococcus aureus* strain ATCC 25923 (*S. aureus*) were used as model microorganisms for the assessment of the antibacterial activity of oxCNDs@GPEIs, parent oxCNDs and GPEIs. *E. coli* bacteria was incubated in LB (Luria-Bertani) medium for 18 h, while *S. aureus* was incubated in tryptic soy broth (TSB) medium for 16 h. Both strains were incubated in a Stuart SI500 orbital shaker (~200 rpm shaking speed, Bibby Scientific Ltd., Staffordshire, UK) at 37 °C in aerobic conditions. Thus, bacteria suspensions with a concentration of ~10^8^ CFU/mL were obtained as established by the measurement of the suspensions’ optical density (OD) at 600 nm using a Cary 100 Conc UV–visible spectrophotometer (Varian Inc., Mulgrave, Australia) and used for the experiments that followed.

#### 2.5.2. Determination of Minimum Inhibitory Concentration and Minimum Bactericidal Concentration

The broth micro-dilution method was used to determine the minimum concentrations of oxCNDs@GPEIs and oxCNDs that were needed to inhibit the growth of *E. coli* and *S. aureus* bacteria and resazurin was used as bacterial health indicator. Taking advantage of the fact that resazurin is a cell-permeable compound and can be metabolically reduced only by viable cells to the fluorescent pink-coloured resorufin, bacterial viability can be measured by monitoring the fluorescence intensity of bacterial cultures [38]. Thus, *E. coli* and *S. aureus* bacteria suspensions at a concentration of ~10^4^ CFU/mL were mixed with various concentrations of GPEIs, oxCNDs and oxCNDs@GPEIs (5–500 μg/mL) in a 96-well plate at 37 °C. After 24 h of incubation time, 5 μL of resazurin solution (6.7 mg/mL) was added to each well and bacteria were further incubated for 4 h. Untreated bacterials were used as the positive control, while LB or TSB media without bacteria were used as the negative control. The fluorescence intensity of the produced resorufin was measured by an Infinite M200 plate reader (Tecan group Ltd., Männedorf, Switzerland, λ_ex_ = 530 nm, λ_em_ = 590 nm). Minimum inhibitory concentrations (MICs) were determined as the lowest concentration at which no fluorescence intensity was recorded, indicating that no bacterial growth occurred.

The minimum bactericidal concentration (MBC) of oxCNDs@GPEIs and oxCNDs was assessed using the colony-counting method. Stock dispersions (5 mg/mL) of each sample were freshly prepared in sterilized water, which were serially diluted in order to obtain dispersions with concentrations ranging from 5 to 750 μg/mL. Then, each dispersion (3 mL) was mixed with the bacteria suspension (10 μL), and incubated on a Stuart SI500 orbital shaker (~200 rpm shaking speed) at 37 °C. After 24h of incubation time, the dispersions were diluted with the appropriate medium, and then 100 μL of each dilution was dispersed into LB agar plates. The LB agar plates were incubated at 37 °C. After 18 h of incubation time for *E. coli* or 16 h for *S. aureus*, the colonies on the plates were counted and compared to the control. For the control experiment, untreated bacteria were used, following the same procedure. All tests were repeated at least three times. The MBC value was considered as the concentration in which a 3-log reduction in the viability of the initial bacterial inoculum had taken place.

#### 2.5.3. Bacteria Morphology (SEM)

The morphology of the *E. coli* bacteria after treatment with oxCNDs@GPEIs was evaluated using scanning electron microscopy (Jeol JSM 7401F Field Emission SEM, JEOL USA Inc., Peabody, MA, USA). Specifically, bacteria were incubated with oxCNDs@GPEIs at their 50% inhibitory concentration (IC50) for 12 h, transferred to a poly(L-lysine)-coated glass cover slip, fixed with 3% glutaraldehyde in sodium cacodylate buffer (100 mM, pH = 7.1), dehydrated using an ethanol gradient (twice of 50%, 70%, 95%, and 100% ethanol for 10 min each), dried, and coated with gold in a sputter coater [39].

### 2.6. Evaluation of Cell Cytotoxicity

Human embryonic kidney HEK293 cells were obtained from the American Type Culture Collection (ATCC, Manassas, VA, USA). The cells were cultured in Dulbecco’s modified Eagle medium high glucose (D-MEM) supplemented with 10% FBS, 100 U/mL penicillin, 100 μg/mL streptomycin solution and 2 mM L-Glutamine. The cells were incubated at 37 °C in a 5% CO_2_ humidified atmosphere and sub-cultured twice a week after detaching with 0.05% (*w*/*v*) trypsin and 0.02% (*w*/*v*) EDTA solution. All treatments were performed in complete medium.

The cytotoxicity of oxCNDs@GPEIs was evaluated using the MTT assay. HEK293 cells were inoculated (10^4^ cells/well) in 96-well plates and left to incubate in complete media for 24 h. Then, cells were treated for 24 h with oxCNDs@GPEIs dispersed in D-MEM, at various concentrations close to MBC values. After this period, the cell medium was removed and 100 μL of MTT solution (10 μg/mL in D-MEM) was added to each well and cells were further cultured for 4 h. Then, the produced formazan crystals were solubilized by adding 100 μL isopropanol to each well and their absorbance was measured at 540 nm using an Infinite M200 microplate reader (Tecan group Ltd., Männedorf, Switzerland). The relative cell viability was determined as percentage compared to untreated cells (control). Blank values measured in wells with isopropanol and no cells were in all cases subtracted.

## 3. Results and Discussion

### 3.1. Synthesis and Characterization of GPEI-Functionalized oxCNDs

Hyperbranched polyethyleneimines (PEIs) with molecular weights of 5000 or 25,000 Da were functionalized at the primary amino groups with guanidinium groups as descripted in our previous publication [35,40]. Subsequently, these GPEI derivatives, which combine the PEI scaffold and guanidinium moieties in the same molecule, were, for the first time, interacted with oxidized carbon nanodisks [28] through covalent and non-covalent bonds. Oxidized nanodisks were used as a safe alternative to GO, since oxCNDs were shown to be a non-toxic material in our previous study [28] and verified in this study (see Section 3.3), unlike GO, which, as mentioned in the literature, exhibits significant toxicity towards eukaryotic cells [31,32,33,34]. In our previous work [35], the strong interaction between GPEIs and oxidized carbon nanotubes (oxCNTs), which occurred through both electrostatic interactions and bidentate hydrogen bonds due to the electronic and geometrical complementarity of the guanidinium moieties of GPEIs with the carboxylate groups of oxCNTs, was demonstrated. On the other hand, other studies [41,42] reported that amino-containing dendritic polymers can be efficiently attached to the GO framework through covalent bonds, which take place between the amino groups of polymers and the epoxy groups of GO without, however, excluding the formation of ionic bonds between the amino groups of polymers with the other oxygen functional groups of GO. Thus, a combination of non-covalent bonds (electrostatic interaction and bidentate hydrogen bonds) between the guanidinium groups of GPEIs and the carboxylate groups of oxCNDs, together with the covalent bonds between the amino groups of GPEIs with the epoxy groups of oxCNDs, took place, leading to the formation of two functional nanomaterials, oxCNDs@GPEI5K and oxCNDs@GPEI25K, which were subsequently characterized by various physico-chemical techniques, such as XPS, FTIR, RAMAN, SEM, TEM, etc.

The successful introduction of GPEIs into the framework of oxCNDs was initially confirmed by X-ray photoelectron spectroscopy, investigating the type of interactions between oxCNDs and GPEIs. Specifically, the emergence of the N1s peaks in the XPS survey spectra of oxCNDs@GPEI5K and oxCNDs@GPEI25K (Appendix A) confirms the presence of GPEI5K and GPEI25K, respectively, as these peaks do not show in the reference survey spectrum of oxCNDs (Appendix A). In Figure 1a and Figure 2a, the deconvoluted C1s photoelectron spectra of GPEI5K and GPEI25K are shown. The spectra reveal four components (peaks) representing the C-C, C-N, and C=N bonds; and C-N^+^ groups, the latter two confirming the successful guanidinylation of the terminal primary amino groups of the parent PEI. This is corroborated by the deconvoluted N1s photoelectron spectra of GPEI5K and GPEI25K (Figure 1b and Figure 2b), which reveal the constituent =N- and -N^+^ groups of guanidinium groups in addition to the -NH- groups present in the backbone of the hyperbranched GPEI [35]. Figure 1c and Figure 2c show the deconvoluted C1s photoelectron spectra of oxCNDs@GPEI5K and oxCNDs@GPEI25K, obtained by fitting them using four components. Both spectra reveal the intensification of the second and third components, located approximately at 286 and 287 eV, respectively, relative to those of oxCNDs (Appendix A), which may be due to the contribution of the C-N and C=N bonds present in GPEI5K and GPEI25K. Furthermore, comparing the deconvoluted N1s photoelectron spectra of oxCNDs@GPEI5K and oxCNDs@GPEI25K (Figure 1d and Figure 2d) with those of GPEI5K and GPEI25K (Figure 1b and Figure 2b), respectively, a sub-electron volt shift in the binding energies of all the N1s components of the oxCNDs@GPEI hybrids, relative to those of the neat GPEIs, can be detected, possibly attributed to strong electrostatic van der Waals interactions as well as to the hydrogen bonding between the oxygen-containing groups of oxCNDs and the amino/guanidinium groups of GPEIs [35,43].

The above-discussed results are also confirmed by FTIR studies. Specifically, in the oxCND spectrum (Appendix A), the broad band centered at 3350 cm^−1^, attributed to the O−H stretching vibrations, as well as the bands at 1710 cm^−1^, 1413 cm^−1^ and 1060 cm^−1^, assigned to the C=O stretching vibrations of the –COOH groups, the O−H bending vibrations and the C–O stretching vibration of alkoxy groups, respectively, verify the presence of hydroxyl and carboxyl groups in the oxCND framework [28]. Additionally, the band at 1221 cm^−1^ is ascribed to C–O stretching of epoxy groups [44]. In the FTIR spectra of both GPEIs (Appendix A), all the characteristic bands, i.e., at 3260 and 3140 cm^−1^ (N−H stretching vibration of primary and secondary amino groups of guanidinium moieties, respectively), 2950 and 2830 cm^−1^ (asymmetrical and symmetrical CH_2_ stretching vibrations, respectively), 1644 and 1619 cm^−1^ (symmetrical and asymmetrical C=N stretching vibrations of guanidinium groups, respectively), 1455 cm^−1^ (bending mode of CH_2_) as well as at 1105 and 1050 cm^−1^ (asymmetrical and symmetrical C-N vibrations), were shown [35]. On the other hand, in the corresponding FTIR spectra of oxCNDs@GPEIs (Appendix A), various differences were observed compared to those of the parent components. Specifically, the band at 1221 cm^−1^ assigned to C–O stretching of epoxy groups disappeared, suggesting that amino groups of GPEIs reacted with epoxy groups of oxCNDs. An analogous reaction has been observed in the case of GO when interacting with amino-rich dendritic polymers in a basic environment [41,42], where the amino groups are involved in the ring-opening reaction of the epoxide groups. Additionally, the C=O stretching band at 1710 cm^−1^ is not detected in the oxCNDs@GPEI spectra, implying that ionic interactions between carboxylate groups and guanidinium moieties took place. Finally, all the other characteristic bands originating from oxCNDs and GPEIs also appeared in the FTIR spectra of GPEI-functionalized oxCNDs, revealing their successful interaction.

The successful introduction of GPEIs in the framework of oxCNDs was also studied using Raman spectroscopy. The Raman spectra of CNDs, oxCNDs and GPEI-functionalized CNDs are presented in Figure 3a. The Raman spectrum of CNDs reveals intense G and G′ (2D) bands and much weaker D and D′ bands. The intense and broad D bands of oxCNDs, oxCNDs@GPEI5K, and oxCNDs@GPEI25K can be attributed to the sp^2^-to-sp^3^ orbital hybridization of the carbon atoms associated with the oxidation of CNDs. This is evidenced by the emergence of the D′, D+G (D+D′), and 2G (2D′) bands, which are characteristic of oxidized graphitic nanomaterials [45,46]. These bands were unveiled upon deconvolution by fitting the Raman spectra of oxCNDs, oxCNDs@GPEI5Κ, and oxCNDs@GPEI25Κ to seven symmetric Lorentzian peaks as shown in Figure 3a. The Raman spectrum of CNDs was fitted to four symmetric Lorentzian peaks corresponding to the D, G, D′, and G′ (2D) bands. The Lorentzian curve fitting parameters for the Raman spectra of CNDs, oxCNDs, oxCNDs@GPEI5Κ, and oxCNDs@GPEI25Κ are presented in Appendix A.

The D-to-G and D′-to-G band intensity ratios ($I_{D}/I_{G}$ and $I_{D^{\prime }}/I_{G}$, respectively) are directly proportional to the defect concentration in graphene [47]. As can be seen in Table 1, $I_{D}/I_{G}$ and $I_{D^{\prime }}/I_{G}$ of oxCNDs are ~10 and 19 times higher compared to CNDs, respectively. Furthermore, the G′-to-G band intensity ratio ($I_{G^{\prime }}/I_{G}$), which is inversely proportional to the defect concentration in graphene [47], decreases by ~10% upon oxidation of CNDs. Therefore, these observations confirm the oxidation of CNDs, also corroborated by the (i) broadening of the G and G′ bands of oxCNDs relative to those of CNDs as revealed by the full width at half-maximum (FWHM) values of their respective G and G′ bands (Appendix A) [48]; and (ii) remarkable decrease in the G′-to-D+G band intensity ratio ($I_{G^{\prime }}/I_{D+G}$) [46].

It is worth noting that the Raman spectra of high-molecular-weight PEIs (Mw ≥ 2.5 kDa) are dominated by strong bands at 2700–3000 cm^−1^ corresponding to the CH_2_ asymmetric and symmetric stretching modes, and weaker bands at 1200–1400 cm^−1^ corresponding to the CH_2_ wagging and twisting mode [49]. The overlapping of these bands with the D, G′ (2D), and D+G (D+D′) bands of oxCNDs makes $I_{D}/I_{G}$, $I_{G^{\prime }}/I_{G}$, and $I_{G^{\prime }}/I_{D+G}$ of oxCNDs@GPEI5Κ and oxCNDs@GPEI25Κ unreliable parameters to quantify the extent of GPEI functionalization of oxCNDs. For this reason, we will solely consider $I_{D^{\prime }}/I_{G}$ in our oxCNDs@GPEI5Κ and oxCNDs@GPEI25Κ sample analysis. As can be seen in Table 1, the $I_{D^{\prime }}/I_{G}$ values of oxCNDs@GPEI5Κ and oxCNDs@GPEI25Κ are ~29% and ~33% lower than those of oxCNDs, respectively. The decrease in defect concentration associated with GPEI functionalization of oxCNDs can be attributed to the reductive nature of parent PEI from which GPEIs were derived [50].

The X-ray diffractograms of oxCNDs, oxCNDs@GPEI5K and oxCNDs@GPEI25K (Figure 3b) reveal the (001) reflex characteristic of oxidized graphitic nanomaterials [51]. The most notable observation is the significant diminishing of the (001) reflex of oxCNDs upon GPEI functionalization, which corresponds to a decrease in the periodicity along the [001] direction. This implies that the graphitic galleries of oxCNDs are partially intercalated with GPEI, which is evidenced by the peak tailing of the (001) reflexes of oxCNDs@GPEI5K and oxCNDs@GPEI25K toward the diffraction angle (2*θ*) range of 7–11°. Deconvolution of the (001) reflexes of oxCNDs@GPEI5K and oxCNDs@GPEI25K was performed using two symmetric Lorentzian peaks as shown in Figure 3b, and the corresponding deconvolution analysis is presented in Table 2. The (001)_B_ peaks of oxCNDs@GPEI5K and oxCNDs@GPEI25K correspond to the domains that are intercalated with GPEI5K and GPEI25K, respectively, whereas the (001)_A_ peaks correspond to the pristine (non-intercalated) domains. The interlayer spacings of the pristine and GPEI-intercalated domains were calculated using Bragg’s equation. As can be seen in Table 2, the intercalation of GPEI5K and GPEI25K increases the interlayer spacing of oxCNDs by 5.63% and 9.72%, respectively. Moreover, the (001)_B_-to-(001)_A_ peak area ratios of oxCNDs@GPEI5K and oxCNDs@GPEI25K reveal an intercalation efficiency of approximately 2:3 (i.e., two GPEI-intercalated domains for every three pristine domains).

More evidence regarding the GPEIs content was obtained by thermogravimetric analysis (TGA). Specifically, in the TGA curve of oxCNDs (Appendix A), a decomposition region between 160 and 250 °C was observed. During this temperature region, a weight loss of ~30% of the initial weight was registered, which is ascribed to the removal of oxygen-containing groups present on the oxCNDs sheets, followed by a temperature region between 250 and 500 °C, where a gradual decomposition of the graphitic framework started (weight loss of about 10%) and completed at about 580 °C. Analogous behavior was observed in the TGA curves of oxCNDs@GPEIs following a slower degradation rate. This was apparently due to the successful attachment of GPEIs, which delays the degradation of oxCNDs, leading to the enhancement of the thermal stability of the final nanomaterials, as their thermal degradation was completed at 610 °C and 640 °C for oxCNDs@GPEI5K and oxCNDs@GPEI25K, respectively, which is significantly higher than that of oxCNDs (580 °C). Similar results were reported when multi-walled carbon nanotubes (oxCNTs) were functionalized with these GPEI derivatives [35], revealing the successful attachment of GPEIs on the sidewalls of oxCNTs.

Moreover, ^1^H NMR spectroscopy was applied for further proof of the presence of GPEIs in oxCNDs@GPEI5K and oxCNDs@GPEI25K as well as for the quantification of the polymer content, using maleic acid as an internal standard [35,37]. Specifically, from the ^1^H NMR spectra of oxCNDs@GPEI5K and oxCNDs@GPEI25K (Appendix A), the presence of GPEIs in the final nanomaterials was established by the peaks at 3.25 ppm and 2.70–2.50 ppm, which are attributed to the protons of α-CH_2_ groups relative to guanidinium groups and the protons of ethylene groups of PEI scaffolds, respectively. Moreover, the determination of GPEI content in the final products was achieved by comparing the integral of the peak at 6.35 ppm, attributed to the protons of methine groups of maleic acid, with that of the peak at 2.70–2.50 ppm. It was found that, on average, 19.2 μmol (0.128 g) GPEI5K and 5.1 μmol (0.170 g) GPEI25K were attached to 1 g of oxCNDs@GPEI5K and oxCNDs@GPEI25K, respectively. Additionally, the quantification of the GPEI content was calculated using the nitrogen mass fraction in the final oxCNDs@GPEIs as the nitrogen signal mainly derives from GPEIs [35,36,52]. As presented in Appendix A, the actual value of the GPEI content in the final oxCNDs@GPEI5K and oxCNDs@GPEI25K was estimated at 13.63% *w/w* and 16.26% *w/w*, respectively, which are in line with the results obtained from ^1^H NMR spectroscopy.

Scanning electron (SEM) and high-resolution transmission electron (HRTEM) microscopies were used to investigate the morphology of oxCNDs after their functionalization with GPEIs. As shown in Figure 4A–C, the parent oxCNDs mainly appear as homogeneous disks with a round shape and a mean diameter of 1–3 μm, while some of them are decorated with carbon nanoparticles with diameters of 30–100 nm or graphene layers appearing as folds partially detached from the disks produced during the oxidation process [28]. Comparison of the SEM images of the parent oxCNDs with those of the functionalized ones (Figure 4) shows that the nanodiscs retain their round shape but more folds appear on their surface due to the partial exfoliation of oxCNDs taking place during the functionalization process, providing further evidence of the successful attachment of GPEIs to oxCNDs.

Analogous results were also found by TEM microscopy, where it was confirmed that the structure of the nanodisks did not significantly change after their modification with GPEIs (Figure 5). Specifically, comparing the TEM images of oxCNDs (Appendix A) with those of oxCNDs@GPEI5K (Figure 5A–E) and oxCNDs@GPEI25K (Figure 5F–J), it was observed that the oxCNDs remain homogeneous after their functionalization, maintaining their original shape and size, while a higher number of graphene layers appearing as folds were detached from the stacking structure of the disks due to their partial exfoliation, revealing the successful attachment of GPEIs to them. Moreover, in TEM images of Figure 5 and Appendix A, carbon nanoparticles (20–50 nm) and/or graphite flakes can be seen on the surface of oxCNDs, which are more clearly visible in Figure 5D,E,J and Appendix A, while in the images of Figure 5D,H, the graphite sheets at the edges of the nanodisks can be clearly observed. Moreover, the presence of GPEIs in oxCNDs@GPEI5K and oxCNDs@GPEI25K was further studied by a combination of high-resolution TEM (HRTEM) and dark-field (HAADF detector) STEM imaging with the corresponding energy-dispersive X-ray (EDX) mapping images. Specifically, information regarding the presence of GPEIs on the surface of oxCNDs was received by determining the spatial distribution of carbon, oxygen and nitrogen from the EDS mapping images of C (K edge), N (K edge) and O (K edge). Comparing the element mapping images of oxCNDs@GPEI5K (Figure 6A) and oxCNDs@GPEI25K (Figure 6B) with those of oxCNDs (Appendix A), it is evident that the GPEIs uniformly decorate the entire surface of the oxCNDs, as nitrogen (red), which is exclusively derived from the GPEIs, is uniformly distributed in the same position where both carbon (green) and oxygen (blue) are located.

### 3.2. Aqueous Colloidal Stability of GPEI-Functionalized oxCNDs

The aqueous colloidal stability of oxCNDs@GPEIs was studied in comparison with that of the parent oxCNDs by visual observation over time. As observed in Figure 7, although both oxCNDs and oxCNDs@GPEIs exhibit excellent aqueous dispersibility, oxCNDs@GPEIs exhibit higher colloidal stability at room temperature over a period of 15 days compared to the parent oxCNDs, which precipitated after settling for 15 days. This behavior can be ascribed to the presence of guanidinium groups on the graphene sheets, which increase the hydrophilicity of oxCNDs, providing improved aqueous stability due to the electrostatic repulsions, also revealing the successful functionalization of oxCNDs with GPEIs. Similar results were also observed when oxidized carbon nanotubes (oxCNTs) were modified with the same GPEI derivatives [35], affording materials that can be effectively dispersed in water due to the presence of the guanidinium groups, providing oxCNTs with aqueous compatibility. Moreover, analogous behavior was mentioned for amido-functionalized GO, which exhibited excellent aqueous stability over a wide time period, better than GO, due to the presence of the active amino groups on the surface of GO [53].

Furthermore, the successful attachment of GPEIs on oxCNDs sheets was confirmed by zeta potential measurements. The *ζ*-potential value of the oxCNDs dispersion was measured at pH = 7.0 and it was found to be −23.2 ± 2.5 mV, due to the negative oxygen-containing groups on the surface of oxCNDs, which is in line with that reported in the literature [28]. On the other hand, the *ζ*-potential values of the oxCNDs@GPEIs dispersions were found to be +31.4 ± 1.0 mV and +34.9 ± 0.8 mV for oxCNDs@GPEI5K and oxCNDs@GPEI25K, respectively. It is obvious that due to the successful modification of the oxCNDs with GPEIs, their surface charge becomes significantly more positive due to the presence of the positively charged guanidinium groups, exhibiting high electrostatic repulsions between the charged nanoparticles, hence their colloidal stability [54].

### 3.3. In Vitro Viability Studies

As the cytocompatibility of the antibacterial agents is a crucial parameter for their applicability in various sectors, the cytotoxicity of the parent oxCNDs and the GPEI-functionalized oxCNDs was initially assessed against the HEK293 normal human kidney cell line. Thus, these cells were incubated with oxCNDs or GPEI-functionalized oxCNDs at various concentrations up to their MIC values for 24 h and then cell viability was evaluated, measuring the mitochondrial redox function of cells by 3-(4,5-dimethylthiazol-2-yl)-2,5 diphenyltetrazolium bromide (MTT) assays. The results are presented in Figure 8 and Appendix A. As can be observed, oxCNDs exhibit only slight toxicity (cell survival > 75%) on HEK293 cells for all tested concentrations (up to 500 μg/mL) after 24 h of incubation time (Appendix A). Analogous results were obtained for both oxCNDs@GPEIs (Figure 8), as they do not exhibit any significant toxicity (cell survival ~80% at the higher tested concentrations 300 μg/mL after a 24 h treatment).

### 3.4. Evaluation of Antibacterial Properties

The antibacterial activity of oxCNDs and oxCNDs@GPEIs was investigated against Gram-negative *E. coli* and Gram-positive *S. aureus* bacteria. Their MIC and MBC values were determined by the broth dilution and colony-counting methods according to M07-A9 and M26-A protocols issued by the Clinical Laboratory Standards Institute (CLSI), respectively [55,56]. The MIC and MBC values of oxCNDs and oxCNDs@GPEIs for *E. coli* and *S. aureus* bacteria are summarized in Table 3. As observed, both oxCNDs@GPEIs exhibit enhanced antibacterial activity compared to the parent oxCNDs, with oxCNDs@GPEI5K being more active than oxCNDs@GPEI25K. Specifically, in both strains, oxCNDs at concentrations up to 750 μg/mL do not exhibit any antibacterial effect in contrast to the structurally analogue GO. This behavior may be attributed to the different shape of GO as it is known that the antibacterial activity of GO is mainly attributed to the stronger interaction of the sheets’ sharp edges with the bacterial membrane, resulting in damage of the cell membrane and finally leading to bacterial death [11,12]. In contrast, oxCNDs cannot follow this mechanism probably due to their disk-like shape, and therefore cannot affect the cell membranes.

On the other hand, both oxCNDs@GPEIs exhibited enhanced inhibitory activity against both tested bacteria strains with MICs and MBCs ranging from 150 to 300 μg/mL and 200 to 300 μg/mL, respectively *(*Table 3). Based on CLSI standards, both hybrid derivatives can be considered as bactericidal material since their MBC/MIC ratio ranges between 1 and 2 [55]. It is obvious that the presence of GPEIs offers antibacterial properties to oxCNDs as GPEIs were found to exhibit efficient antibacterial activity. Specifically, the GPEI5K’s MIC values were 100 and 200 μg/mL for *S. aureus* and *E. coli*, respectively, while for GPEI25K, an MIC value of 100 μg/mL was recorded for both strains. These findings are in line with those in the literature where it has been noted that various polymers functionalized with guanidinium groups exhibited enhanced antibacterial properties [57,58]. This behavior has been attributed to the enhanced electrostatic attraction between the cationic guanidinium moieties and anionic phosphate and carboxylate groups located on the surface of bacteria, leading to disruption of the bacterial membrane, subsequent leakage of the intracellular components and eventually to bacterial death [59]. Moreover, these guanidinium derivatives have been studied as molecular transporters that can effectively penetrate though cellular membranes and translocate into cells [60]. There, they can efficiently interact with cytoplasmic constituents such as proteins and genes, resulting in alteration of bacterial metabolism and finally bacterial death [61]. Similar enhancement in the antibacterial activity of GO was observed when GO was functionalized with polyxamethylene guanidine hydrochloride [19] or chitosan and polyxamethylene guanidine hydrochloride [16]. It was found that the MIC value of GO against *E. coli* was reduced from ~250 μg/mL to 20–30 μg/mL. However, in both cases, the antibacterial properties of the hybrid materials are better than those of oxCNDs@GPEIs and neither of these materials had been investigated for their cytotoxicity. Taking into consideration that GO shows remarkable toxicity from concentrations as low as 85 μg/mL (~50% cell viability) [34,62], there is a great possibility that these GO derivatives are also toxic. Indeed, GO-PEI has been reported to have an MIC value against *E. coli* and *S. aureus* of 8 μg/mL [20], while in another work, the same derivative was found to be highly toxic even at a concentration of 1.6 μg/mL [62]. It is known that in order to be applicable as an efficient antibacterial agent, a material should exhibit simultaneously good antibacterial activity and low toxicity. Hence, despite its excellent antibacterial properties, GO-PEI cannot be used as an antibacterial agent due to its high toxicity. On the other hand, oxCNDs@GPEIs fulfill these criteria as they exhibit good antibacterial activity simultaneously with low toxicity (cell viability > 80% at MBC values i.e., 200–300 μg/mL).

Another crucial parameter related to the enhanced antibacterial activity of oxCNDs@GPEIs is their improved aqueous dispersion compared to oxCNDs, which can facilitate their interaction with bacteria, resulting in greater cell damage. It is known that well-dispersed carbon-based materials, such as carbon nanotubes [36] or graphene oxide [16], can strongly interact with bacteria though direct contact, resulting in extended membrane damage.

Furthermore, oxCNDs@GPEIs showed more pronounced activity against *S. aureus* than against *E. coli*. This finding could be related to the structural differences in their cellular envelope, known to affect cellular properties, particularly responses to external stresses, such as antibiotics [63]. Due to these differences, the complex Gram-negative bacterial cell envelope is tougher to penetrate by various molecules or even widely used antibiotics, compared to that of Gram-positive bacteria. Thus, the latter are more sensitive to various antibacterial agents [64].

The difference in the toxicity against bacteria and eukaryotic cells could be tentatively attributed to the different membrane potential of bacteria and cells, strongly related to the different chemical composition of their membranes. It is known that the cellular membrane potential (Δ*Ψ*) of eukaryotic cells ranges between 30 mV and 60 mV [65], while the membrane potential of bacteria ranges between 100 mV and 150 mV [66]. Thus, due to their high membrane potential, cationic species may be moved across the lipid bilayers electrophoretically, as exemplified by the Nernst equation, correlating the membrane potential with the ratio of the concentration of ions between the inner and the outer media of any membrane [67]. Therefore, the positively charged oxCNDs@GPEIs are expected to be internalized more efficiently in bacteria than eukaryotic cells.

The surface morphology of *E. coli* after 12 h treatment at 37 °C with oxCNDs@GPEIs was investigated by scanning electron microscopy (SEM). SEM images of the control (untreated bacteria) and bacteria treated with oxCNDs@GPEIs at ½ MIC are shown in Figure 9. As observed, the untreated bacteria (Figure 9A,E) seem intact with a smooth surface, while the treated bacteria appear to have lost their cellular integrity as their surface is more rough and some of their cell walls and membranes have been ruptured, resulting in leakage of the intracellular components and apparently bacterial death (Figure 9B–D,F–H). This result could be attributed to the strong interaction of the guanidinium groups located on the surface of carbon nanodisks with the negatively charged components of the outer leaflet of the bacterial cytoplasmic membrane and cell walls through multiple interactions, including electrostatic and hydrogen bonding, which affect the membrane function and integrity, causing membrane permeabilization and precipitation of the bacterial cytoplasm [68,69]. Additionally, it is known that the antibacterial activity of polycationic compounds and especially the guanidinylated polymers can be attributed to their ability to penetrate bacterial membranes easily, interacting with the divalent cations, which are responsible for maintaining membrane integrity [58].

## 4. Conclusions

In this study, for the first time, oxidized carbon nanodisks, as a safe alternative material to graphene oxide, were functionalized with guanidinylated hyperbranched polyethyleneimine derivatives of 5000 and 25,000 Da molecular weight via covalent and non-covalent interactions, producing two novel hybrid derivatives, i.e., oxCNDs@GPEI5K and oxCNDs@GPEI25K, with approximate polymer loading levels of 13–16% *w/w*. Subsequently, these hybrid derivatives were characterized by a variety of physicochemical techniques to confirm their chemical structure. Specifically, the successful attachment of GPEIs to oxCNDs was established by FTIR, Raman, XPS, XRD, TGA, ζ-potential measurements, ^1^H NMR and elemental analysis, while the homogenous anchoring of GPEIs on the surface of oxCNDs was verified by SEM and TEM studies. Additionally, oxCNDs@GPEIs showed improved aqueous stability compared to the parent oxCNDs, due to the presence of guanidinium groups on their surface, which increase the hydrophilicity and electrostatic repulsion of oxCNDs. To assess the antibacterial activity of the as-prepared oxCNDs@GPEIs, Gram-negative *E. coli* and Gram-positive *S. aureus* bacteria were used. It was found that both hybrids exhibited enhanced antibacterial activity against both test organisms, with oxCNDs@GPEI5K being more active than oxCNDs@GPEI25K. In particular, the obtained MIC and MBC values were much lower (150–300 μg/mL and 200–300 μg/mL of oxCNDs@GPEIs, respectively) than those of oxCNDs (>750 μg/mL), revealing that the presence of GPEIs in the final hybrids offers enhanced antibacterial activity. SEM images revealed that probably due to their polycationic character, oxCNDs@GPEIs strongly interact with negatively charged components of the outer leaflet of the bacterial cytoplasmic membrane and cell walls through multiple interactions, including electrostatic and hydrogen bonding, leading to cell envelope damage and precipitation of the bacterial cytoplasm and finally to cell lysis. Moreover, oxCNDs@GPEIs showed minimal cytotoxicity on human normal cells HEK293, indicating that these hybrid nanomaterials have great potential to be used as safe and efficient antibacterial agents.

## Figures and Tables

**Figure 1 nanomaterials-14-00677-f001:**
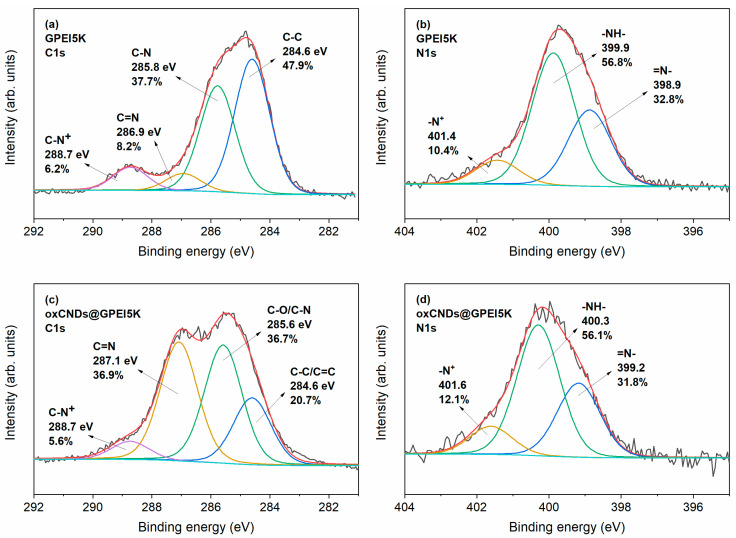
Deconvoluted high-resolution core-level C1s (**a**,**c**) and N1s (**b**,**d**) photoelectron spectra of GPEI5K and oxCNDs@GPEI5K; (**a**,**c**) color legend: C-C/C=C (blue), C-O/C-N (green), C=N (dark yellow), and C-N^+^ (purple); (**b**,**d**) color legend: =N- (blue), -NH- (green), and -N^+^ (dark yellow).

**Figure 2 nanomaterials-14-00677-f002:**
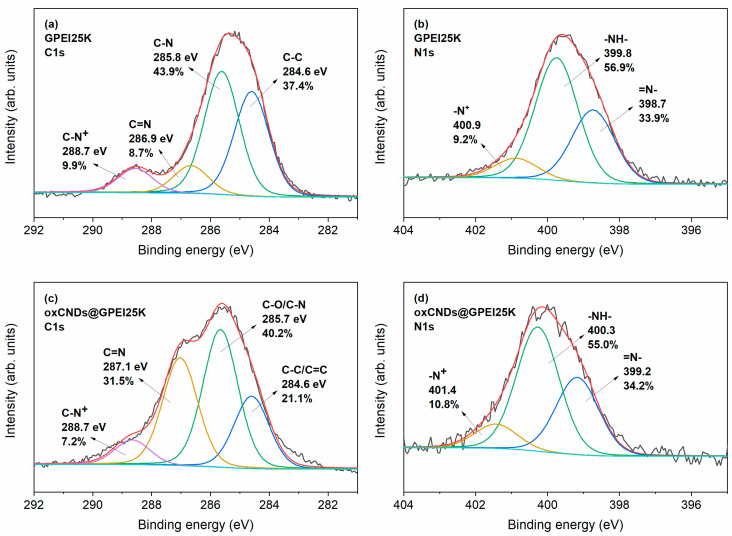
Deconvoluted high-resolution core-level C1s (**a**,**c**) and N1s (**b**,**d**) photoelectron spectra of GPEI25K and oxCNDs@GPEI25K; (a,c) color legend: C-C/C=C (blue), C-O/C-N (green), C=N (dark yellow), and C-N^+^ (purple); (b,d) color legend: =N- (blue), -NH- (green), and -N^+^ (dark yellow).

**Figure 3 nanomaterials-14-00677-f003:**
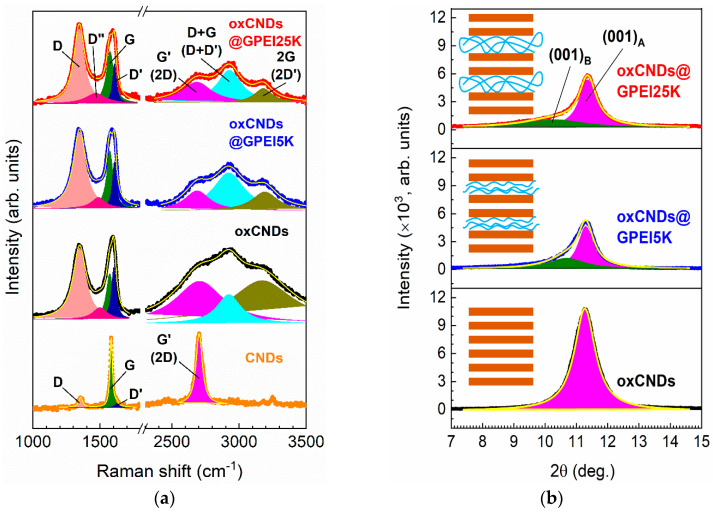
(**a**) Deconvoluted Raman spectra of CNDs, oxCNDs, and GPEI-functionalized oxCNDs. All the bands were fit to Lorentzian profiles. (**b**) Lorentzian curve fitting of the (001) reflexes of oxCNDs, oxCNDs@GPEI5K, and oxCNDs@GPEI25K. Insets: The corresponding schematic representations before and after the partial intercalation of oxCNDs with GPEI5K and GPEI25K; (**a**) color legend for the bands: D (light red), D″ (pink), G (green), D′ (navy blue), G′ (magenta), D+G (cyan), and 2G (dark yellow); (**b**) color legend for the components of the (001) reflexes: non-intercalated (magenta) and GPEI-intercalated (green) domains.

**Figure 4 nanomaterials-14-00677-f004:**
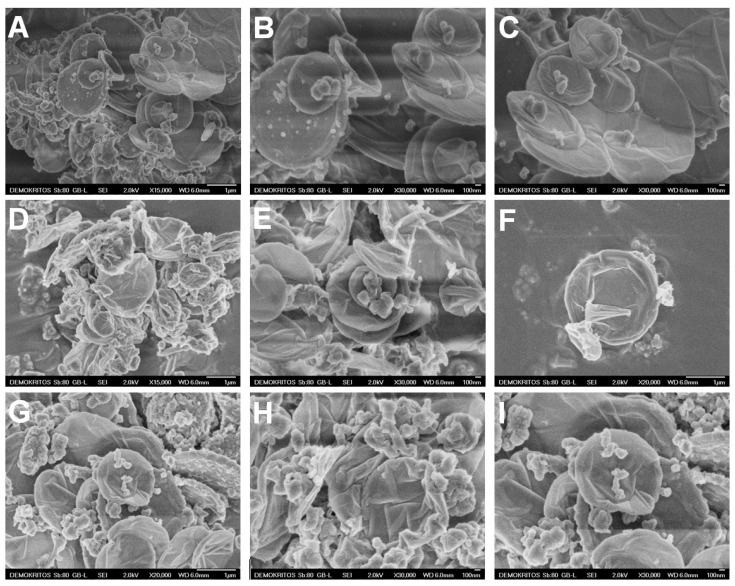
SEM images of oxCNDs (**A**–**C**), oxCNDs@GPEI5K (**D**–**F**) and oxCNDs@GPEI25K (**G**–**I**).

**Figure 5 nanomaterials-14-00677-f005:**
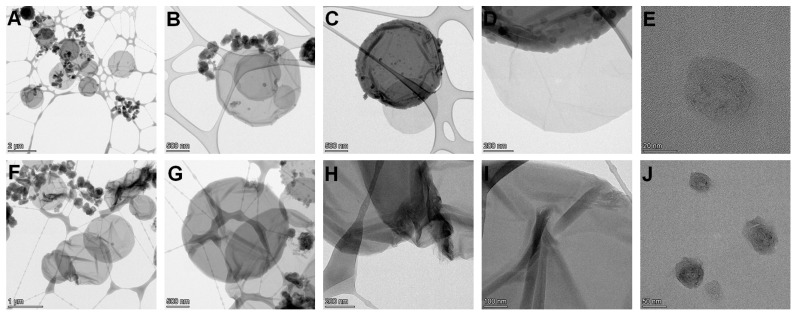
TEM images of oxCNDs@GPEI5K (**A**–**E**) and oxCNDs@GPEI25K (**F**–**J**).

**Figure 6 nanomaterials-14-00677-f006:**
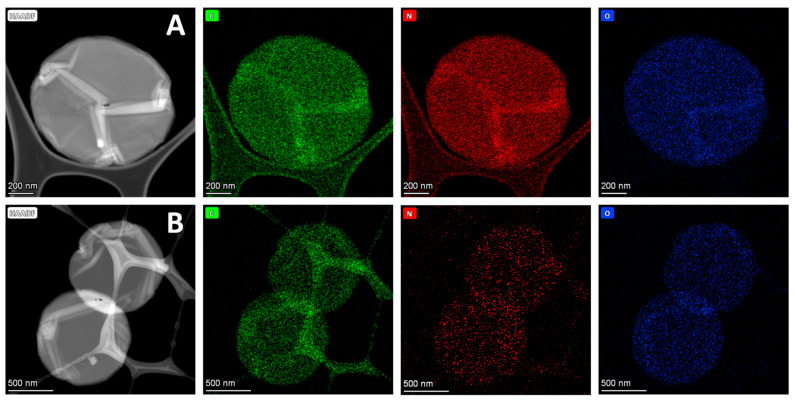
Scanning/transmission electron microscopy high-angle annular dark field images presenting the morphology of oxCNDs@GPEI5K (**A**) and oxCNDs@GPEI25K (**B**) and the corresponding EDS elemental mapping images of C (K edge), N (K edge) and O (K edge).

**Figure 7 nanomaterials-14-00677-f007:**
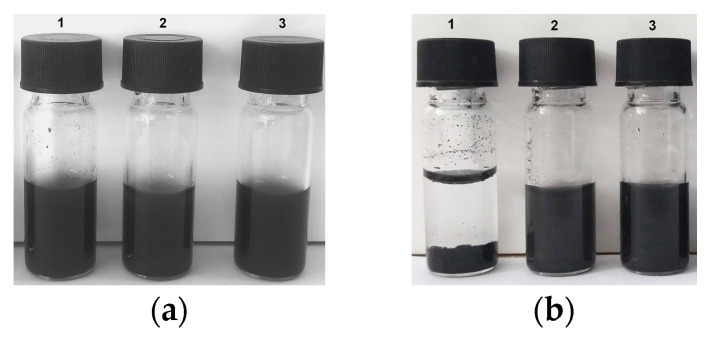
Digital images of oxCNDs (1), oxCNDs@GPEI5K (2) and oxCNDs@GPEI25K (3) aqueous dispersions at a concentration of 1 mg/mL, immediately after sonication (**a**) and after standing still for 15 days (**b**).

**Figure 8 nanomaterials-14-00677-f008:**
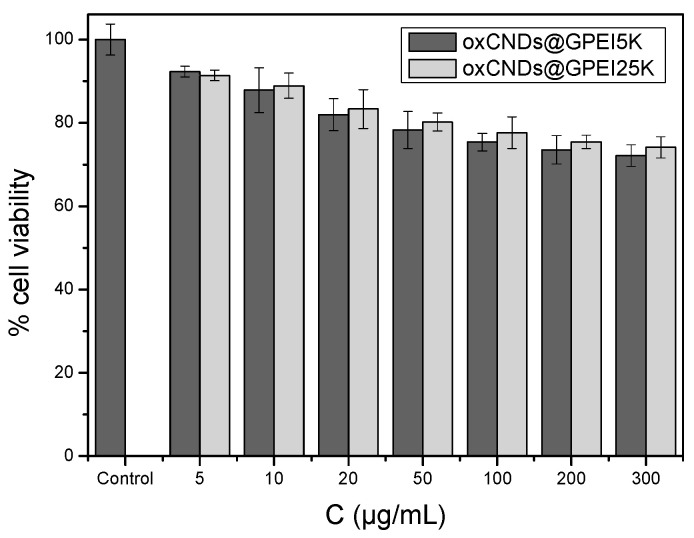
Comparative toxicities of GPEI-functionalized oxCNDs on human embryonic kidney HEK293 cells following incubation at various concentrations for 24 h as determined by MTT assays. Data are expressed as mean ± SD of six independent values obtained from at least three independent experiments.

**Figure 9 nanomaterials-14-00677-f009:**
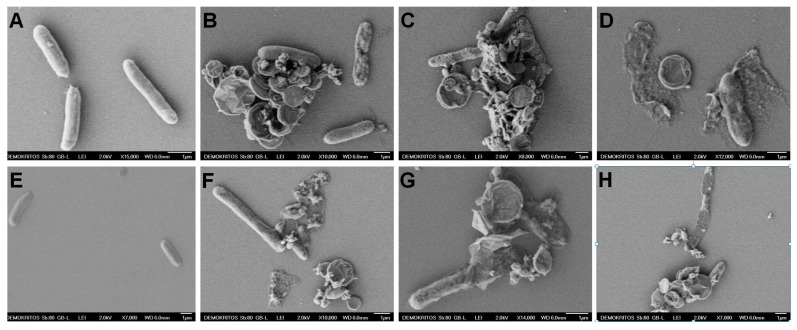
SEM images of *E. coli* bacteria: untreated cells (**A**,**E**) and cells after 12 h incubation time at 37 °C with oxCNDs@GPEI5K (**B**–**D**) or oxCNDs@GPEI25K (**F**–**H**) at ½ MIC.

**Table 1 nanomaterials-14-00677-t001:** Calculated Raman band intensity ratios for CNDs, oxCNDs, and GPEI-functionalized oxCNDs. For CNDs, the value of $I_{G^{\prime }}/I_{D+G}$ was assumed due to the absence of a D+G (D+D′) band.

Sample	$I_{D}/I_{G}$	$I_{D^{\prime }}/I_{G}$	$I_{G^{\prime }}/I_{G}$	$I_{G^{\prime }}/I_{D+G}$
CNDs	0.15	0.06	1.03	∞
oxCNDs	1.56	1.14	0.93	1.46
oxCNDs@GPEI5K	-	0.81	-	-
oxCNDs@GPEI25K	-	0.76	-	-

**Table 2 nanomaterials-14-00677-t002:** Deconvolution analysis of the (001) reflexes of oxCNDs, oxCNDs@GPEI5K, and oxCNDs@GPEI25K. *d* = Interlayer spacing; *A* = area.

Sample	(001)_A_ Peak			(001)_B_ Peak		
	2*θ* (deg.)	*d* (Å)	*A* (%)	2*θ* (deg.)	*d* (Å)	*A* (%)
oxCNDs	11.3	7.82	100	-	-	-
oxCNDs@GPEI5K	11.3	7.82	62.6	10.7	8.26	37.4
oxCNDs@GPEI25K	11.4	7.76	61.7	10.3	8.58	38.3

**Table 3 nanomaterials-14-00677-t003:** MIC and MBC values of oxCNDs and oxCNDs@GPEIs against *E. coli* and *S. aureus* bacteria.

Samples	*E. coli*		*S. aureus*	
	MIC (μg/mL)	MBC (μg/mL)	MIC (μg/mL)	MBC (μg/mL)
oxCNDs	>750	>750	>750	750
oxCNDs@GPEI5K	250	280	150	200
oxCNDs@GPEI25K	300	300	200	250

## Data Availability

The data presented in this study are available upon request from the corresponding author.

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
