# Peer review of "Carbon Nanodisks Decorated with Guanidinylated Hyperbranched Polyethyleneimine Derivatives as Efficient Antibacterial Agents"

_nanomaterials, 2024, doi:10.3390/nano14080677_

Round 1

Reviewer 1 Report

Comments and Suggestions for Authors

In this article submitted by Kyriaki-Marina Lyra, et al., and entitled "Carbon nanodisks decorated with guanidinylated hyper branched polyethyleneimine derivatives as efficient antibacterial agents ", the acid-treated carbon nanodisks (ox-18 CNDs) were functionalized with guanidinylated hyper-19 branched polyethyleneimine derivatives through both covalent and non-covalent interactions, resulting in the synthesis of an efficient antibacterial agents. The paper is very interesting and meaningful, and the English writing is also of high quality, but there are little issues that the author needs to address before it can be considered for publication.

(1) Line 125, the specific purity of the solvents should be provided.

(2) There are some format errors of the citation of references in the manuscript, such as reference of 15 and 19, please check and revise it.

(3) Line 185, the cultivation temperature of the test microorganisms should be provided.

(4) Line 216, “S. aureus” is replaced with “S. aureus”

(5)In the table 3, “St. aureus” should be replaced with “S. aureus”.

(6) In this study, the GPEI-functionalized oxCNDs showed enhanced antimicrobial activity without significant hemolytic activity. The author should add the discussion why AC7 showed the selective antimicrobial activity over hemolytic activity in the section of “3.4. In vitro viability studies”

Author Response

We would like to thank you for the time dedicated to reading and commenting on our manuscript. We have tried to address all the issues raised, by changing the manuscript accordingly (see all our changes highlighted in yellow) and our responses are presented in the attached file.

Reviewer 2 Report

Comments and Suggestions for Authors

In this manuscript, guanidinylated hyperbranched polyethyleneimine derivatives were used to decorate oxidized carbon nanodisks to create two antibacterial materials. This work is generally systematic. The authors provide a detailed characterization of the structures of the prepared samples, and the antibacterial activity of these samples is significant. However, the authors fail to demonstrate the advantages and design rationality of the synthesized antibacterial materials. Since polyethyleneimine and guanidine have already been used for the modification of graphene oxide yet, my main concern about this manuscript is the necessity of using oxidized carbon nanodisks as substrate. As demonstrated by the authors, the antibacterial activity of oxidized carbon nanodisks is lower than that of graphene oxide due to the absence of sharp edges. It is not clear whether this shortcoming would impair the performance of the final product. Without comparing the results to existing literature, the audience cannot assess whether the materials in this manuscript are superior to the reported graphene oxide-based antibacterial materials, particularly those with polyethyleneimine and guanidine decorations. Therefore, in my opinion, this manuscript requires at least a major revision before it can be published.

Author Response

We would like to thank you for the time dedicated to reading and commenting on our manuscript. We have tried to address all the issues raised, by including additional work and changing the manuscript accordingly (see all our changes highlighted in yellow) and our responses are presented in the attached file.

Round 2

Reviewer 2 Report

Comments and Suggestions for Authors

All the concerns have been properly addressed, and this mansucript could be published now.